# Recursive Abstractive Processing for Retrieval in Dynamic Datasets

## Abstract

Recent retrieval-augmented models enhance basic methods by building a hierarchical structure over retrieved text chunks through recursive embedding, clustering, and summarization. The most relevant information is then retrieved from both the original text and generated summaries. However, such approaches face limitations with dynamic datasets, where adding or removing documents over time complicates the updating of hierarchical representations formed through clustering. We propose a new algorithm to efficiently maintain the recursive-abstractive tree structure in dynamic datasets, without compromising performance. Additionally, we introduce a novel post-retrieval method that applies query-focused recursive abstractive processing to substantially improve context quality. Our method overcomes the limitations of other approaches by functioning as a black-box post-retrieval layer compatible with any retrieval algorithm. Both algorithms are validated through extensive experiments on real-world datasets, demonstrating their effectiveness in handling dynamic data and improving retrieval performance.

## 1 Introduction

Large Language Models (LLMs) have established themselves as powerful tools across a wide range of natural language processing (NLP) tasks, thanks to their ability to store vast amounts of factual knowledge within their parameters (Petroni et al., 2019; Jiang et al., 2020). These models can be further fine-tuned to specialize in specific tasks (Roberts et al., 2020), making them highly versatile. However, a key limitation of LLMs lies in their static nature: as the world evolves and new information emerges, the knowledge encoded within an LLM can quickly become outdated. A promising alternative to relying solely on parametric knowledge is retrieval augmentation (Gao et al., 2023). This approach involves the use of external retrieval systems to supply relevant information in real-time. Instead of encoding all knowledge directly into the model, large text corpora are indexed, segmented into manageable chunks, and dynamically retrieved as needed (Lewis et al., 2020; Gao et al., 2023). Retrieval-augmented methods not only improve model accuracy but also offer a practical solution for maintaining performance as knowledge evolves over time.

However, retrieval-augmented approaches also have limitations. Many existing methods only retrieve short, specific chunks as context, which restricts the model's ability to answer questions requiring a broader understanding of the text. To address this, RAPTOR was introduced (Sarthi et al., 2024). It recursively embeds, clusters, and summarizes text chunks, enabling the retrieval of relevant information from both original document chunks and generated summaries. Yet, RAPTOR introduces new challenges, especially with dynamic datasets where documents are frequently added or removed. The clustering component makes the tree structure sensitive to these updates, requiring a full re-computation of the tree after each change, which is computationally expensive.

To address these limitations, we introduce **adRAP** (adaptive Recursive Abstractive Processing), an algorithm designed to efficiently update RAPTOR's recursive-abstractive structure as new documents are added or removed. By incrementally adjusting the structure, adRAP avoids full re-computation, preserving retrieval performance while significantly reducing computational overhead. Furthermore, both RAPTOR and adRAP introduce memory overhead and require periodic maintenance when used with dynamic datasets. As an alternative, we propose **postQFRAP**, a post-retrieval method that applies query-focused recursive abstractive processing as a black-box layer, as illustrated in Figure 1. This post-processing method integrates seamlessly into any retrieval pipeline

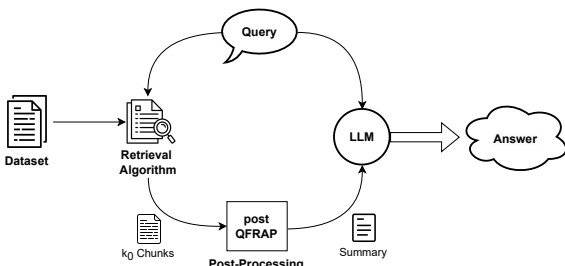

Figure 1: **Retrieval pipeline with postQFRAP**: we first retrieve from a dataset $k_0$ chunks relevant to the query, then we build a query-focused recursive-abstractive tree on those chunks. Finally, we summarize the contents of the root layer of that tree to get the context that is passed to the LLM.

while significantly enhancing the quality of the retrieved context. For example, naïve RAG (Gao et al., 2023) can serve as the underlying model since it processes documents independently, allowing easy addition or removal of documents. Moreover, by initially retrieving enough documents, questions requiring a broader understanding can be answered by passing the generated summary to the LLM, rather than passing all potentially relevant documents, thus mitigating challenges like limited context window size and information loss in large contexts (Liu et al., 2024; Yu et al., 2024). Through extensive experiments on real-world datasets, we demonstrate that adRAP provides a good approximation of the RAPTOR tree, while postQFRAP effectively enhances retrieval quality.

## 2 RELATED WORK

**Retrieval Algorithms**  In the context of LLMs, retrieval-augmented generation (RAG) involves retrieving relevant information from external sources and appending it to the LLM's context alongside the original query (Ram et al., 2023). Naïve RAG methods (Gao et al., 2023) address this challenge by converting documents into text, splitting it into chunks, and embedding these chunks in a vector space where semantically similar chunks are mapped to nearby vectors. The query is similarly embedded, and the $k$-nearest vectors are retrieved to augment the LLM's context. However, segmenting text into contiguous chunks may fail to capture its full semantic richness, and retrieving overly granular segments can overlook key information (Gao et al., 2023).

The recursive-abstractive summarization method by Wu et al. (2021) addresses this issue by breaking down tasks to summarize smaller text segments, which are then integrated to form summaries of larger sections. While effective at capturing broader themes, it may miss finer details. RAPTOR (Sarthi et al., 2024) improves on this technique by recursively grouping and summarizing similar chunks, retaining both summaries and initial chunks. This approach captures a representation of the text at multiple levels of detail while preserving inter-dependencies within the text.

**Post-Retrieval Algorithms**  To optimize retrieval algorithms, post-retrieval strategies are commonly employed. These include re-ranking retrieved chunks and compressing the context (Gao et al., 2023), as large contexts fed directly into LLMs often result in information loss, particularly in the middle sections (Liu et al., 2024; Yu et al., 2024). The closest approach to our setting is the query-focused summarization algorithm by Zhang et al. (2024). They retrieve relevant documents which they also summarize using a prompt designed to extract key information before generating the summary. The latter is then passed as context to the LLM. In contrast, we construct a hierarchical tree over the retrieved documents, allowing us to recursively filter noise by focusing on smaller, manageable chunks at each step. This yields a more refined and relevant summary.

## 3 PRELIMINARIES

### 3.1 CLUSTERING WITH GAUSSIAN MIXTURE MODELS (GMMS)

Gaussian Mixture Models assume that data points are generated from a mixture of multiple Gaussian distributions. They have two key advantages: they allow non-isotropic Gaussians, enabling varied

cluster shapes and orientations, and they support soft clustering, where a data point can belong to multiple clusters. Let $K$ represent the number of clusters, and $x_1, \ldots, x_n$ be the data points. Each cluster $k$ is defined by its mean $\mu_k$, covariance matrix $\Sigma_k$, and mixture weight $\pi_k$, which represents the prior probability of a data point belonging to cluster $k$. The probability density function (PDF) for a data point $x$ is given by $p(x) = \sum_{k=1}^{K} \pi_k \mathcal{N}(x|\mu_k, \Sigma_k)$ where $\mathcal{N}(x_i|\mu_k, \Sigma_k)$ is PDF of a multivariate normal distribution with mean $\mu_k$ and covariance $\Sigma_k$. The cluster parameters are learned by maximizing the log-likelihood using the Expectation-Maximization (EM) algorithm (Moon, 1996), which iterates the two following steps until convergence, i.e., when the change in log-likelihood between consecutive iterations becomes negligibly small.

**Expectation step:** Compute the posterior probability (responsibility) that the $k$-th Gaussian component generated the data point $x_i$:

$$\gamma(z_{ik}) = \frac{\pi_k \mathcal{N}(x_i|\mu_k, \Sigma_k)}{\sum_{j=1}^{K} \pi_j \mathcal{N}(x_i|\mu_j, \Sigma_j)}, \tag{1}$$

**Maximization step:** Update the parameters $\pi_k$, $\mu_k$, and $\Sigma_k$ by maximizing the expected log-likelihood given the responsibilities:

$$\pi_k = \frac{1}{n} \sum_{i=1}^{n} \gamma(z_{ik}), \quad \mu_k = \frac{\sum_{i=1}^{n} \gamma(z_{ik}) x_i}{\sum_{i=1}^{n} \gamma(z_{ik})}, \quad \Sigma_k = \frac{\sum_{i=1}^{n} \gamma(z_{ik})(x_i - \mu_k)(x_i - \mu_k)^\top}{\sum_{i=1}^{n} \gamma(z_{ik})} \tag{2}$$

## 3.2 Dimensionality Reduction with UMAP

Clustering algorithms often struggle with the curse of dimensionality, where data becomes sparse, and distances between points lose distinction in high dimensions. To address this, Uniform Manifold Approximation and Projection (UMAP) (McInnes et al., 2018) reduces the dimensionality of embeddings, significantly enhancing clustering performance (Allaoui et al., 2020). UMAP learns a low-dimensional representation that preserves both local and global structures, with the key parameter *n_neighbors* controlling the trade-off between local and global structure preservation.

## 3.3 Recursive-Abstractive Tree Construction

The process of building the recursive-abstractive tree is outlined first, as it is key to understanding our new algorithms. The construction, based on Sarthi et al. (2024) with minor adjustments, consists of four steps: dataset chunking, clustering, summarizing, and recursive construction.

**Dataset Chunking**   Given a dataset, the first step is to divide the text into sentences using the NLTK Punkt Sentence Tokenizer[1]. These sentences are then grouped into chunks of up to 250 tokens, with a 50-token overlap between consecutive chunks, resulting in chunks of up to 300 tokens. To maintain coherence, sentences are kept intact between chunks: if a sentence exceeds 250 tokens, it is included in the next chunk. Sentences longer than 250 tokens are split at punctuation marks. Token counts are determined using the `cl100k_base` tokenizer from the `tiktoken`[2] library.

Note that we use 250 tokens with a 50-token overlap instead of the 100-token chunks used by (Sarthi et al., 2024). Preliminary experiments (not included in this work) suggest that the larger chunk size with overlap improves output quality.

**Clustering**   The goal is to group $n$ chunks $c_1, \ldots, c_n$ into $k$ clusters $C_1, \ldots, C_k$, where $k$ is to be determined. Clustering is performed on the embeddings, not the raw text. So, using an encoder model, embeddings $v_1, \ldots, v_n$ are generated for the chunks. Then, dimensionality reduction is performed using UMAP, followed by clustering with Gaussian Mixture Models (GMMs). This process is repeated twice, varying UMAP's *n_neighbors* parameter to create a hierarchical clustering, an approach shown to be effective for this task (Sarthi et al., 2024).

First, *n_neighbors* is set to $\sqrt{n}$, generating 10-dimensional embeddings $v_1^g, \ldots, v_n^g$. GMMs are then applied, yielding global clusters $C_1^g, \ldots, C_{k_g}^g$. Next, refinement occurs within each global cluster.

---

[1] https://www.nltk.org/api/nltk.tokenize.punkt.html
[2] https://github.com/openai/tiktoken

UMAP is applied with *n_neighbors* set to 10, resulting in reduced embeddings $v_1^l, \ldots, v_m^l$, where $m$ is the size of the current global cluster. GMMs are then used to form local clusters $C_1^l, \ldots, C_{k_l}^l$. The final clustering is the union of all local clusters. To determine $k_g$, values from 1 to $\max(50, \sqrt{n})$ are evaluated, and we select the value that minimizes the Bayesian Information Criterion (BIC) (Schwarz, 1978). A similar approach is used to determine $k_l$.

**Summarizing**    After clustering, a large language model generates summaries for each cluster, providing a concise overview of the content. The summary length is limited to 1,000 tokens to ensure the summaries remain manageable. The specific prompt used for summarization is provided in the appendix (Table 4).

**Recursive Construction**    The clustering and summarization process is repeated recursively to obtain a multi-layered representation of the dataset. This approach is outlined in Algorithm 1.

---

**Algorithm 1** Recursive-Abstractive Tree Construction

---

1: **Input:** Dataset
2: **Output:** Recursive-Abstractive Tree
3: Chunk the dataset, initializing the leaf nodes as these chunks.
4: **while** the top layer contains more than 10 nodes **and** there are fewer than 5 layers **do**
5:      Compute embeddings for the nodes in the top layer.
6:      Apply the two-step clustering process to group these nodes.
7:      Generate a summary for each cluster.
8:      Form a new layer with one new node per cluster.
9: **end while**

---

## 3.4    RETRIEVING DOCUMENTS

Given a query and a tree constructed over a relevant dataset, the goal is to retrieve $k$ documents that are helpful in answering the query. Sarthi et al. (2024) compared tree-based retrieval with a collapsed-tree approach, where all nodes are considered simultaneously. The latter performed better, and is the method used in our experiments.

In the collapsed-tree approach, the tree is flattened, and the $k$ most similar documents are retrieved using cosine similarity on the embeddings. This method can be seen as augmenting the dataset with document summaries, followed by applying naïve RAG to the expanded dataset. The pseudo-code for the retrieval algorithm is provided in Appendix A.

## 4    ADRAP: ADAPTIVE RECURSIVE-ABSTRACTIVE PROCESSING

### 4.1    OVERVIEW

The problem we are addressing can be formally described as follows. Let $T_0$ represent a recursive-abstractive tree built on an initial dataset $D_0$. Given an updated dataset $D = D_0 \cup D_1$, where $|D_0| \gg |D_1|$, let $T$ be the tree constructed over $D$. The goal is to efficiently update $T_0$ to approximate $T$ without fully recomputing the tree on $D$.

To achieve this, UMAP is used to reduce the dimensionality of the new documents, which are then assigned to clusters, potentially updating the existing clustering. We first examine these components individually, then explain how they are combined to create a dynamic data structure.

### 4.2    ADAPTIVE UMAP

Let $d \in D_1$ be a new document with embedding $v$. The first step is to reduce the dimensionality of $v$ to 10. To do this, we find the $n\_neighbors$ nearest neighbors of $v$ in the original high-dimensional space and interpolate their positions in the previously learned low-dimensional embedding to obtain the reduced embedding $v'$. This preserves the local relationships of $v$ with its neighbors, maintaining the structure learned during fitting. Given $|D_0| \gg |D_1|$, we assume this property holds for all new

documents. This process requires storing the fitted UMAP models (both global and local) with our tree. We use the `UMAP-learn`[3] library for UMAP fitting and dynamic transformations.

### 4.3 ADAPTIVE GMM

A key component of the recursive-abstractive tree construction (Algorithm 1) is its clustering algorithm, which poses challenges when handling dynamic datasets. Given a fitted Gaussian Mixture Model (GMM) $\mathcal{I}$ with $K$ clusters defined by their means $\{\mu_k\}_{k=1}^K$, covariance matrices $\{\Sigma_k\}_{k=1}^K$, and mixing coefficients $\{\pi_k\}_{k=1}^K$, and given points $\{x_i\}_{i=1}^n$ assigned to these clusters, the goal is to assign a new point $x_{n+1}$ to one or more clusters. This may involve updating the clustering structure or introducing new clusters. While prior work addresses online GMMs (Song & Wang, 2005; Declercq & Piater, 2008; Zhang & Scordilis, 2008), our setting differs in that we start with a GMM fit on a dataset, we have access to all the points and we want to minimize the number of updated clusters, as each update requires multiple re-generated summaries.

First, assume $n$ is large, i.e., many points have already been clustered. Given a new point $x_{n+1}$, we compute its posterior probability $\gamma(z_{n+1,k})$ for $k \in [K]$, and approximate the maximization step by updating the parameters with the new point's contribution as follows:

$$\mu_k \leftarrow \frac{n\pi_k\mu_k + \gamma(z_{n+1,k})x_{n+1}}{n\pi_k + \gamma(z_{n+1,k})}, \quad \Sigma_k \leftarrow \frac{n\pi_k\Sigma_k + \gamma(z_{n+1,k})(x_{n+1} - \mu_k)(x_{n+1} - \mu_k)^\top}{n\pi_k + \gamma(z_{n+1,k})}$$

$$\pi_k \leftarrow \frac{n\pi_k + \gamma(z_{n+1,k})}{n+1} \tag{3}$$

The updated parameters match those from applying Equation 2 to the points $\{x_i\}_{i=1}^{n+1}$. After updating the cluster parameters, we recompute the posterior for $x_{n+1}$ and assign it to clusters $\{k : \gamma(z_{n+1,k}) > 0.1\}$, without affecting other point assignments. Although this remains an approximation, it has been shown to be an effective way to incrementally fit a GMM (Neal & Hinton, 1998). The update is efficient, as its time is independent of $n$, with only a few clusters being updated (those assigned to $x_{n+1}$). When $n$ is small, we perform full EM steps instead of updating using only the new point, as the smaller number of clusters makes this affordable. This also yields more significant improvements, as clusterings with fewer points are more sensitive to new data.

At this stage, an issue may arise as the number of clusters $k$ remains fixed, whether we use approximate or full EM steps. If points are repeatedly added to the same cluster, it may grow too large, causing a node at layer 1 to resemble one at layer 3, which undermines the hierarchical structure. To address large clusters, we attempt to split them, thereby increasing $k$. The splitting approach varies with $n$. For large $n$, we focus on the large clusters independently of other clusters. We attempt to subdivide these large clusters by applying a GMM to them with $k' = 1, 2, 3$ subclusters, and we select the best model according to the Bayesian Information Criterion (BIC). This method has a runtime independent of $n$, and at most 3 clusters are updated or created. For small $n$, we explore larger values of $k$ and fit a new GMM from scratch, selecting the $k$ that optimizes the BIC.

We summarize these ideas in Algorithm 2. The parameter $\tau_n$ controls the trade-off between quality and computation time, determining whether to perform full or approximate EM updates based on $n$. Similarly, $\tau_c$ sets the cluster size threshold for triggering a potential split.

### 4.4 adRAP ALGORITHM

The process starts with an initial tree $T_0$ and a new data chunk $d \in D_1$. First, $d$'s embedding $v$ is computed, and a corresponding leaf node is created in $T_0$. The first layer above the leaves (call it layer 1) is then updated to account for the new node.

To do so, the global reduced embedding $v_g$ is derived from $v$ using the global UMAP model and assigned to the most probable cluster in the global clustering. Since the global clustering includes all $|D_0|$ nodes, it is considered stable and no dynamic adjustments are made. Next, we focus on the global cluster to which $v_g$ was assigned, applying the local UMAP model to compute a reduced local embedding $v_l$. The local clustering is updated using Algorithm 2, potentially creating new nodes.

---

[3]`https://umap-learn.readthedocs.io/en/latest/`

---

**Algorithm 2** Adaptive Clustering

---

1: **Input:** GMM Instance $\mathcal{I}$ with $n$ points, new point $x_{n+1}$, thresholds $\tau_n, \tau_c$
2: **Output:** Updated Instance $\mathcal{I}'$
3: **if** $n \leq \tau_n$ **then**
4:     Perform full EM steps until convergence.
5:     Let $c$ be the number of clusters with more than $\tau_c$ points.
6:     Fit GMM instances with $K$ up to $K + c$ clusters, keeping the best one with respect to BIC.
7: **else**
8:     Perform a maximization step using $x_{n+1}$'s contribution.
9:     Assign $x_{n+1}$ to clusters $\{k : \gamma(z_{n+1,k}) > 0.1\}$.
10:     **if** some cluster $k$ contains more than $\tau_c$ points **then**
11:         Fit a GMM on cluster $k$ to get a sub-clustering with at most 3 clusters.
12:     **end if**
13: **end if**

---

In $T_0$, nodes with updated children regenerate their summaries and recompute embeddings, with updates propagated to their ancestors (up to five levels). If new clusters are created at layer $i$, this procedure is recursively applied at layer $i + 1$. By design, only a few clusters are affected, minimizing the need for summary re-computation. To illustrate this, we compare in Appendix H.2 the runtime and number of generated summaries between adRAP and a full re-computation of the tree. Moreover, the pseudo-code of adRAP is presented in Appendix A.

Though we focused on adding documents, the algorithm easily handles deletions by removing the chunk from the tree and recomputing summaries for its ancestors. For frequent deletions, one can either recompute the local clustering by trying smaller values for $K$ or leave the clusters unchanged.

## 5   POSTQFRAP: POST-RETRIEVAL QUERY-FOCUSED RECURSIVE-ABSTRACTIVE PROCESSING

### 5.1   MOTIVATION

Maintaining adRAP is costly, as it requires updating clusters and summaries with each new document. Moreover, when many documents are added, the entire tree has to be recomputed to maintain solution quality. Integrating this system poses significant development challenges, and companies with established retrieval algorithms may be hesitant to adopt a completely new system.

To address this, we propose a modified version of the recursive-abstractive tree as a black-box post-retrieval solution that can be integrated with retrieval algorithms handling dynamic datasets (e.g., naïve RAG). This approach enhances the initial construction by incorporating query-focused summaries, improving the context relevance of the output.

### 5.2   POSTQFRAP ALGORITHM

Let $\mathcal{R}$ be a retrieval algorithm that takes as input an integer $k \in \mathbb{N}^+$ and a query $q$, and returns $k$ documents relevant to the query. A simple example is the naïve RAG algorithm (Gao et al., 2023).

We augment $\mathcal{R}$ as follows. First, we retrieve $k_0$ documents without imposing a token limit. Then, we apply a query-focused version of Algorithm 1 to these $k_0$ documents to build a recursive-abstractive tree. The key modification is using query-focused summarization (see prompt in Appendix, Table 5). Since the tree is constructed to answer $q$, summarizing information relevant to $q$ ensures that key details are preserved while recursively filtering out irrelevant content. Additionally, we modify the clustering to rely solely on local embeddings, as retrieving $k_0$ documents already serves as a global filtering step. In other words, we assume the retrieved documents belong to the same global cluster. We demonstrate in Appendix B that using the simpler one-step clustering preserves the quality of the generated context compared to the two-step approach.

Finally, a summarization step is applied to all nodes at the last layer of the tree, instead of using a top-$k$ retrieval approach, to reduce redundancy in the results. This process is detailed in Algorithm 3.

---

**Algorithm 3** postQFRAP Algorithm

---

1: **Input:** Retrieval Algorithm $\mathcal{R}$, Initial Number of Chunks $k_0$, Query $q$, Token Threshold $\tau$
2: **Output:** Final summary with at most $\tau$ tokens
3: Retrieve $k_0$ documents using $\mathcal{R}(k_0, q)$.
4: Construct a tree $T$ on the $k_0$ documents using Algorithm 1 with query-focused summarization and one-step clustering.
5: Generate a final query-focused summary from the content of the top layer of $T$, using at most $\tau$ tokens.
6: Return the final summary.

---

### 5.3 KEY PROPERTIES

postQFRAP can be seamlessly integrated as a black-box solution with any retrieval algorithm that handles dynamic datasets. A prime example of the latter is the naïve RAG algorithm, where adding documents is easy: chunk the new documents, embed each chunk, and add them to the vector database. Removing documents is equally simple—just delete them from the database.

Moreover, by recursively applying query-focused summarization, postQFRAP continuously extracts information relevant to answering the question. Then, the final summarization step removes redundancy and serves as a last denoising phase, producing a highly relevant and coherent context. It is important to note that increasing the hyperparameter $k_0$ enables the model to handle broader questions without expanding the generated context size, though it increases inference time.

Furthermore, postQFRAP avoids relying on a summarization model with a large context length due to its recursive structure, which focuses on small chunks at each step. This enables the use of a distilled model for greater inference efficiency (e.g., the abstractive compressor of Xu et al. (2023)).

## 6 EXPERIMENTS

### 6.1 DATASETS

We evaluate our methods on three question-answering datasets: *MultiHop*, *QASPER*, and *QuALITY*.

*MultiHop* consists of news articles published between 2013 and 2023 (Tang & Yang, 2024). Although the original questions focus on retrieving and reasoning across multiple documents, they primarily target explicit fact retrieval. To create more challenging questions requiring a broader understanding, we construct a RAPTOR tree on the dataset, sample chunks/summaries from the tree, and ask an LLM to generate questions based on those chunks. Details are provided in Appendix H.3.

*QASPER* consists of 1,585 NLP papers with associated questions (Dasigi et al., 2021). Each question seeks information from the full text and is written by an NLP practitioner who has only seen the title and abstract. For our experiments, we use the first 300 questions and their relevant papers. To make each question context-independent, we include the paper's name in the question (e.g. instead of asking "What are the observed results?", we ask "In paper X, what are the observed results?")

*QuALITY* consists of multiple-choice questions paired with context passages averaging 5,000 tokens (Pang et al., 2022). This exceeds the size of the context generated by the retrieval algorithms in our experiments. We also select the first 300 questions along with their corresponding context passages.

We present the results of an additional dataset in Appendix H.4. Moreover, dataset sizes and an analysis of the recursive-abstractive trees constructed for each dataset are provided in Appendix H.6.

### 6.2 METRICS

A key factor in the evaluation is the prompt used for the Question-Answering model. To focus on the effectiveness of retrieval algorithms, we instruct the model to rely solely on the provided context. The full prompt is in the Appendix (Table 7).

To evaluate our algorithms, we use two methods: a rating-based evaluation, providing a score for each model independently, and a head-to-head comparison. Since the model is restricted to using

only the retrieved context, measuring faithfulness is unnecessary. Instead, we focus on ensuring the context provides sufficient information to answer the question. Thus, we compute the proportion of **answered questions** and measure **context relevance** (Es et al., 2023). The latter acts as context precision, while the former is analogous to context recall, as it checks whether the necessary chunks are retrieved. However, we avoid using context recall directly, as it is difficult to formally define with summarized chunks.

Some generated answers may lack coherence, either in their internal structure or in relation to the question. Providing summarized content as context may help the model generate more coherent responses. To evaluate this, we introduce a novel metric called **Human Coherence Rating**, which prompts an LLM to assess whether an answer is coherent and resembles one that could plausibly be generated by a human expert. The specific prompt used for this evaluation is shown in the appendix (Table 6), with a qualitative analysis of the metric provided in Appendix C.

To gain deeper insights into our algorithms, we conduct a head-to-head evaluation. Given our focus on questions requiring a global understanding, we adopt the evaluation metrics from Edge et al. (2024), which assess **comprehensiveness**, **diversity**, **empowerment**, and **directness**. For each comparison, the evaluator LLM is given the question, a prompt describing the target metric, and two answers. The LLM evaluates which answer is superior or if it is a tie, providing a rationale for its decision. To mitigate position bias (Zheng et al., 2024), the evaluation is repeated for each pair of answers with their positions swapped. If the same answer wins both trials, it is declared the winner; otherwise, the result is a tie.

We also conduct a qualitative analysis of postQFRAP, detailed in Appendix D.

### 6.3 HYPERPARAMETER SELECTION

A key hyperparameter to consider is the context size, or equivalently, the number of documents to retrieve. Sarthi et al. (2024) evaluated different context lengths on a subset of the QASPER dataset, finding that 2,000 output tokens yielded the best results. Based on this, we set the output context size to 2,000 tokens for all algorithms unless stated otherwise.

To select $k_0$ for the postQFRAP algorithm, we compare different values on two validation datasets. We choose $k_0 = 20$, as larger values increase computational complexity without significant quality gains, while smaller values substantially reduce context relevance. Details of this study are provided in Appendix E.

### 6.4 BASELINES

We compare the adRAP algorithm against Naïve RAG, RAPTOR, and a greedy variant of adRAP, which assigns each new point to its most probable cluster without updating the GMM fit. To compute adRAP, we first construct a full tree using 70% of the dataset. The remaining 30% is added using the adRAP algorithm (Section 4.4) where we set $\tau_c = 11$ and $\tau_n = \max(100, \sqrt{|D_0|})$ for Algorithm 2. The choice of $\tau_c$ is based on the average cluster size in the full RAPTOR tree, which is always less than 10 (see appendix, Table 12). For the greedy variant, a similar procedure is used. To simulate a challenging scenario, we remove the last 30% of documents instead of random sampling.

We compare postQFRAP with other post-retrieval methods: no processing (naïve RAG) with $k = 7, 20$ retrieved documents, one-shot summarization, re-ranking, and postRAP. One-shot summarization uses the controller from Zhang et al. (2024) to directly generate 2,000 tokens (see prompt in Appendix, Table 8). For re-ranking, we use the `ms-marco-MiniLM-L-12-v2`[4] cross-encoder from HuggingFace, retrieving 20 documents via naïve RAG, then re-ranking them to keep the top 7. Finally, postRAP is a variant of postQFRAP without query-focused summarization which retrieves the top-$k$ most similar chunks from the tree built on the $k_0$ chunks.

We also tried adding query expansion (Jagerman et al., 2023) to our postQFRAP algorithm, but this barely affected the results. So, we report the details of those experiments in Appendix F.

---

[4]https://huggingface.co/cross-encoder/ms-marco-MiniLM-L-12-v2

We use OpenAI's `text-embedding-3-large`[5] for embeddings and `gpt-4o-mini-2024-07-18`[6] for all LLM tasks. All retrieval algorithms use a chunk size of 300 tokens with a 50-token overlap. To account for the non-determinism of LLM evaluators, we repeat each experiment three times, reporting the average and standard error.

## 6.5 RESULTS

Figure 2 shows that adRAP's performance is generally on par with RAPTOR across most metrics, with the exception of context relevance, where adRAP falls short by at least 3%. However, adRAP outperforms both the naïve RAG and the greedy algorithm, particularly in the QuALITY dataset. These findings are further corroborated by the head-to-head evaluations in Figures 3, 4, and 5. Notably, in the QuALITY dataset, adRAP exceeds RAPTOR in metrics such as comprehensiveness, diversity, and empowerment, despite its lower performance in context relevance. On the other hand, adRAP underperforms compared to RAPTOR in the MultiHop and QASPER datasets.

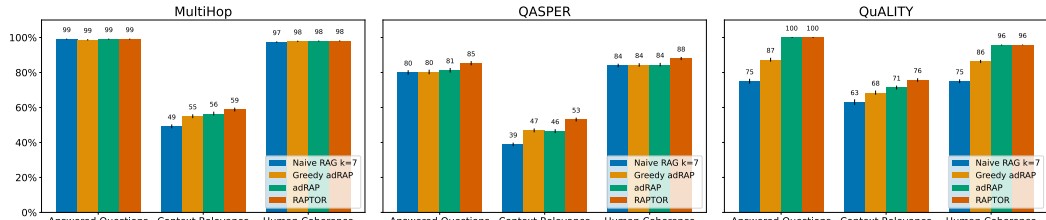

Figure 2: Evaluation of adRAP (Section 4.4) on 3 datasets.

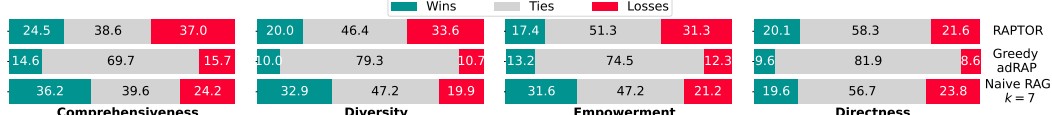

Figure 3: Percentage of Wins, Ties and Losses for adRAP vs other algorithms on MultiHop.

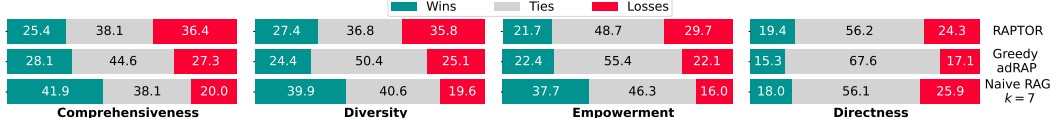

Figure 4: Percentage of Wins, Ties and Losses for adRAP vs other algorithms on QASPER.

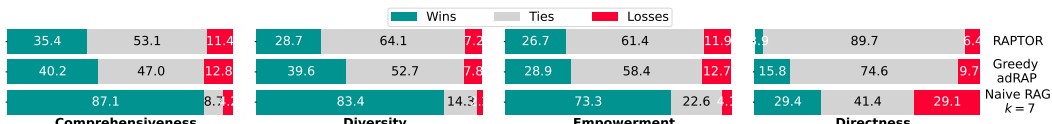

Figure 5: Percentage of Wins, Ties and Losses for adRAP vs other algorithms on QuALITY.

Figure 6 shows that algorithms with query-focused summarization consistently outperforms other approaches across all metrics. While one-shot summarization scores slightly higher in answered questions and human coherence, postQFRAP excels in context relevance, demonstrating the effectiveness of recursive summarization in filtering noise from input chunks. The superiority of postQFRAP as a post-retrieval algorithm becomes apparent in head-to-head evaluations. As shown in Figures 7, 8, and 9, postQFRAP excels in comprehensiveness, diversity, and empowerment. The lower directness scores are expected, as directness often contrasts with these qualities, as noted by Edge et al. (2024). Overall, postQFRAP's recursive extraction produces a more diverse, comprehensive, and empowering context, enhancing the quality of the final answer.

---

[5]https://platform.openai.com/docs/guides/embeddings
[6]https://platform.openai.com/docs/models/gpt-4o-mini

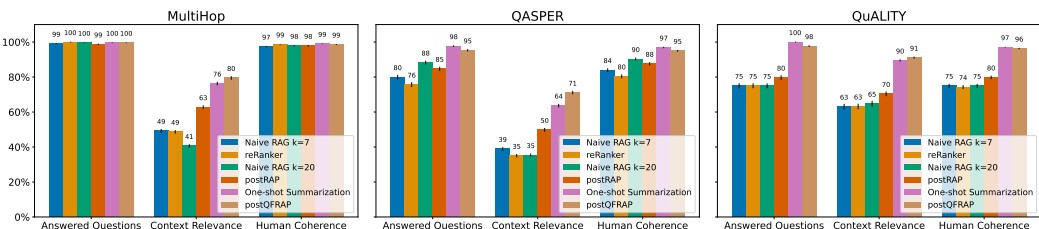

Figure 6: Evaluation of postQFRAP (Algorithm 3) on 3 datasets.

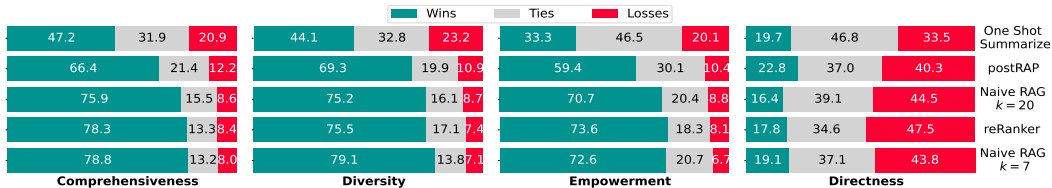

Figure 7: Percentage of Wins, Ties and Losses for postQFRAP vs other algorithms on MultiHop.

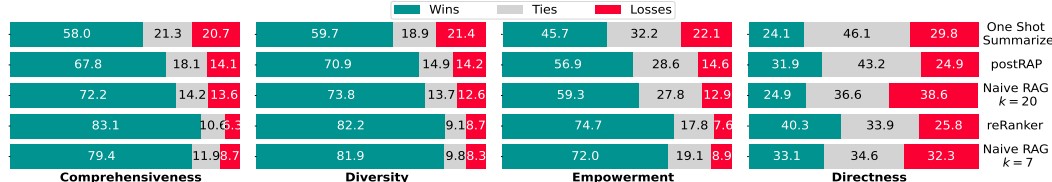

Figure 8: Percentage of Wins, Ties and Losses for postQFRAP vs other algorithms on QASPER.

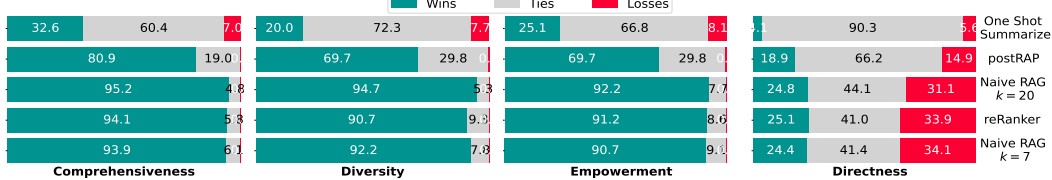

Figure 9: Percentage of Wins, Ties and Losses for postQFRAP vs other algorithms on QuALITY.

# 7 LIMITATIONS

While adRAP is more efficient than repeatedly recomputing the full RAPTOR tree for dynamic datasets, it introduces some overhead. It requires extra memory to store multiple UMAP and GMM models and adds complexity to the retrieval pipeline, as it must be triggered when new documents are added. Additionally, a full tree recomputation may still be needed if a large volume of new documents is introduced, increasing implementation effort. With postQFRAP, generated summaries can make it harder to trace original sources. Additionally, multiple summarization calls are required during inference, although this follows the current trend of shifting more computational workload to inference time, as seen with OpenAI's o1 model (OpenAI, 2024; Brown et al., 2024).

# 8 CONCLUSION

In this paper, we introduced adRAP, an adaptive extension of the RAPTOR algorithm, designed to efficiently approximate clustering when documents are added or removed. Our experiments show that adRAP performs comparably to RAPTOR, making it a viable solution for dynamic datasets.

We also presented postQFRAP, a novel post-retrieval algorithm that applies query-focused, recursive-abstractive processing to refine large contexts. By filtering out irrelevant information, postQFRAP produces highly relevant summaries. Our results demonstrate that postQFRAP consistently outperforms traditional methods, proving its effectiveness for post-retrieval processing.

## 9 REPRODUCIBILITY STATEMENT

**Language Model Used** Open AI's `gpt-4o-mini-2024-07-18`[7] is used for both question answering and summarization in all our experiments. Open AI's `text-embedding-3-large`[8] is used to generate embeddings.

**Prompts** All used prompts are presented in Appendix G.

**Hyperparameters** All hyperparameters and model configurations used in the experiments are clearly detailed in Sections 6.3 and 6.4.

**Datasets** All four datasets used in our experiments are publicly available: MultiHop, NarrativeQA, QuALITY, and QASPER. Details of the preprocessing steps are provided in Appendix H.5.

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

## A    PSEUDOCODES

---
**Algorithm 4** Querying the Recursive-Abstractive Tree
---
1: **Input:** Query $q$, Recursive-Abstractive Tree $T$, Integer $k$, Token Threshold $\tau$
2: **Output:** List of retrieved documents
3: Compute the embedding $v$ for the query $q$.
4: Calculate the cosine similarity between $v$ and the embeddings of all nodes in $T$.
5: Select the $k$ most similar nodes, sorted by decreasing similarity.
6: Add the nodes' content to the output in order, stopping if the token threshold is reached.

---

---
**Algorithm 5** adRAP Algorithm
---
1: **Input:** Tree $T_0$ with GMM and UMAP models, new document $d$
2: **Output:** Updated tree $T$
3: Compute the embedding $v$ of document $d$ using the appropriate model.
4: Create a leaf node in $T_0$ for $(d, v)$.
5: Compute the global reduced embedding $v_g$ of $v$ using the global UMAP model.
6: Assign $v_g$ to the most probable cluster in the global clustering, denoted $C_g^*$.
7: Compute the local reduced embedding $v_l$ of $v$ using the local UMAP model of $C_g^*$.
8: Update the local clustering of $C_g^*$ using the online GMM procedure (Algorithm 2).
9: **for all** clusters that changed **do**
10:       Regenerate the tree summary.
11:       Recompute the embedding for the cluster node.
12:       Repeat the process for its ancestors in the tree.
13: **end for**
14: **for all** newly created clusters **do**
15:       Recur to the next layer (treating it as the leaf layer) and repeat the process.
16: **end for**

---

## B    COMPARING ONE- VS TWO-STEP CLUSTERING FOR POSTQFRAP

We compare postQFRAP, which uses a two-step hierarchical clustering algorithm as described in Section 3.3, with a one-step approach that only applies local clustering (i.e., setting the UMAP parameter $n\_neighbors$ to 10 and using GMMs once).

As shown in Figures 10,the results across all four datasets indicate that the difference between one-step and two-step clustering is minimal. Furthermore, Figure 11 demonstrates that one-step clustering performs better in QASPER and QuALITY, worse in NarrativeQA, and is comparable to two-step clustering in MultiHop. Overall, the differences in performance between the two algorithms are minor. Based on these observations, we adopt the simpler and more efficient one-step clustering method in our algorithm.

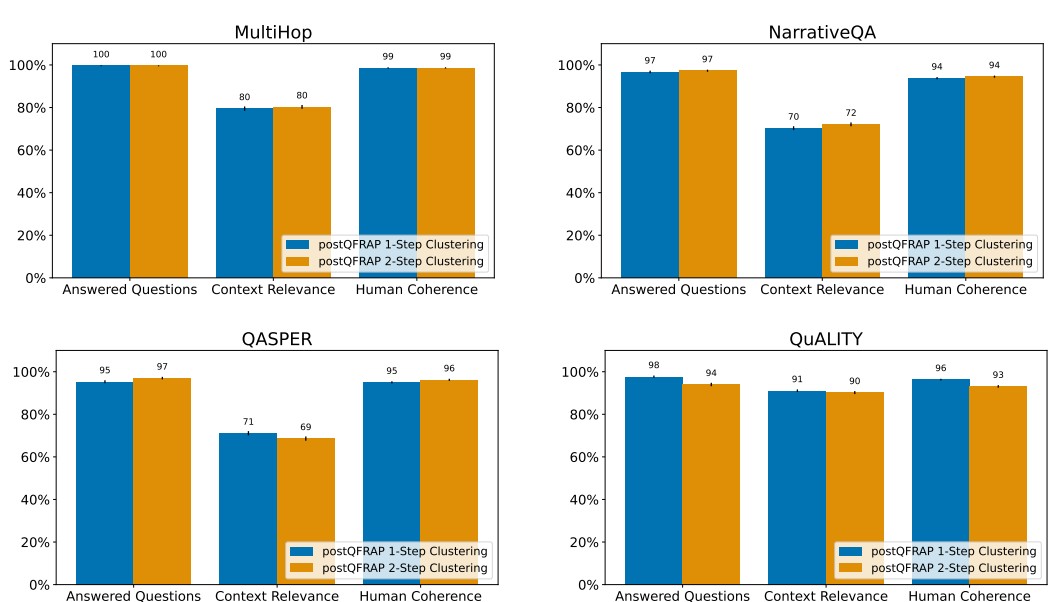

Figure 10: Comparison of postQFRAP with one-step clustering and postQFRAP with two-step clustering on 4 datasets.

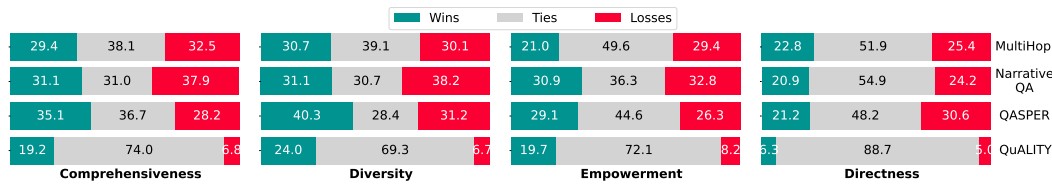

Figure 11: Percentage of Wins, Ties and Losses for postQFRAP with one-step clustering (green) vs postQFRAP with two-step clustering (red) on 4 datasets.

## C  QUALITATIVE ANALYSIS OF HUMAN COHERENCE RATING

To qualitatively analyze the newly proposed Human Coherence Rating, we consider an example of a question from the MultiHop dataset where using naïve RAG led to a low rating. The results are summarized in Table 1.

Table 1: Comparison of Human Coherence Ratings for generated answers to a MultiHop dataset question, evaluating the performance of Naïve RAG versus postQFRAP models.

| **Question: What injury has prevented Michail Antonio from participating in the match against Canada?** | | |
|---|---|---|
| **Naïve RAG** | Generated Answer | Michail Antonio has been sidelined due to a gutting ACL injury sustained during the match against Canada. He is now beginning a lengthy rehabilitation process. |
| | Human Coherence Rating | 0.4 |
| | Rating Explanation | The provided answer lacks coherence due to an error in the terminology used; an ACL injury (anterior cruciate ligament injury) is described as "gutting," which is an inappropriate term in this context. Additionally, the phrase "sustained during the first leg of the match against Canada" suggests that the injury occurred during the match itself, which could be misleading. Furthermore, the transition to discussing the rehabilitation process feels abrupt and does not smoothly connect with the explanation of the injury. |
| **postQFRAP** | Generated Answer | Michail Antonio was unable to participate in the match against Canada due to a significant injury that has been identified as a possible hamstring issue. There are also reports suggesting that his injury may be more severe than initially thought, with some sources indicating it could potentially be an ACL (anterior cruciate ligament) injury. This type of injury typically requires a lengthy rehabilitation process, which could keep him out of action for an extended period. |
| | Human Coherence Rating | 1.0 |
| | Rating Explanation | The provided answer is coherent and presents a clear response to the question about Michail Antonio's injury. The information is logically structured, beginning with the immediate reason for his absence (a significant injury) and then elaborating on the nature and severity of that injury (possible hamstring issue and potential ACL injury). The flow of ideas is smooth, with each sentence building upon the previous one, maintaining relevance to the question throughout. |

## D  QUALITATIVE ANALYSIS OF POSTQFRAP

To qualitatively analyze the postQFRAP algorithm, we examine a question from the MultiHop dataset. Table 2 presents the contexts and answers generated by postQFRAP and naïve RAG. The context produced by naïve RAG is scattered and often irrelevant to the question, causing the QA model to fail in providing an answer. In contrast, postQFRAP generates a coherent and highly relevant context, resulting in significant portions of the final answer being directly extracted from this context.

Table 2: Comparison of contexts and answers generated for a MultiHop question using Naïve RAG and postQFRAP.

| Question: What measures has the U.K. Judicial Office implemented to ensure the responsible use of AI in the judicial system? | | |
|---|---|---|
| **Naïve RAG** | Generated Context | [...] "the use of AI in sentencing, parole and probation, pretrial release and detention, risk assessments, surveillance, crime forecasting and predictive policing, and forensic analysis." [...] prioritizing federal support to accelerate development of privacy-preserving techniques [...] cryptographic tools that preserve individuals' privacy [...] "We have also established a governance team to put them into action by conducting ethical reviews of new systems, avoiding bias and incorporating privacy, security and safety." [...] "Citizens will have a right to launch complaints about AI systems and receive explanations about decisions based on high-risk AI systems that impact their rights." [...] The use of remote biometric identification technology in public places by law enforcement has not been completely banned [...] the parliament said negotiators had agreed on a series of safeguards and narrow exceptions to limit use of technologies such as facial recognition. [...] with uses limited to a "strictly defined" lists of crime. [...] |
| | Generated Answer | No information is provided in the sources. |
| **postQFRAP** | Generated Context | The U.K. Judicial Office has recently issued guidance that allows judges to utilize AI tools, including ChatGPT, in their legal duties and court rulings. This guidance is designed to promote the responsible and ethical use of AI within the judicial system. [...] Judges are encouraged to critically assess the outputs generated by AI tools and to consider how these outputs may reflect or perpetuate existing biases. [...] By fostering a culture of critical engagement with AI, the Judicial Office aims to mitigate the risks associated with automated decision-making. [...] the guidance underscores the necessity of upholding privacy standards when using AI in court proceedings. Judges are reminded to protect sensitive information and ensure that the use of AI does not compromise the confidentiality of legal processes. [...] the retrieved documents do not detail specific measures or actions taken by the Judicial Office beyond this guidance. [...] they primarily focus on broader discussions surrounding AI regulations in other contexts, such as the European Union's AI Act and various U.S. executive orders related to AI. [...] These regulations do not directly pertain to the U.K. Judicial Office's measures. [...] |
| | Generated Answer | The U.K. Judicial Office has implemented guidance that promotes the responsible and ethical use of AI tools, including ChatGPT, within the judicial system. Key measures outlined in this guidance include: 1. **Awareness of Potential Bias**: Judges are encouraged to critically assess the outputs generated by AI tools. [...] 2. **Upholding Privacy Standards**: The guidance emphasizes the necessity of protecting sensitive information during court proceedings. [...] 3. **Promoting Critical Engagement**: By fostering a culture of critical engagement with AI, the Judicial Office aims to mitigate the risks associated with automated decision-making. [...] |

# E   SELECTION OF $k_0$ FOR POSTQFRAP

To determine the optimal value of $k_0$ for our main experiments, we evaluated five different values, $k_0 \in \{10, 20, 40, 60, 80\}$, by comparing the performance on 100 questions from the validation

sets of the NarrativeQA and QASPER datasets. It is important to emphasize that these questions are entirely distinct from the data used in our main experiments. As in our primary experiments, we limited the final summary size to 2,000 tokens and considered three metrics: the number of answered questions, context relevance, and human coherence. To account for the non-deterministic nature of the LLM evaluators, we repeated the evaluation process three times and reported the average performance along with the standard error in Figure 12.

As expected, context relevance increases with higher $k_0$ values, as including more documents allows additional potentially relevant content to be retained while ensuring irrelevant content is excluded from the summaries.

In both validation sets, we observe that the fraction of answered questions and human coherence peaks at $k_0 = 20$. Although $k_0 \in \{60, 80\}$ provides higher context relevance compared to $k_0 = 20$, we select $k_0 = 20$ for our experiments as it optimizes the number of answered questions and human coherence without sacrificing too much context relevance. Additionally, this choice ensures a more efficient post-retrieval process compared to larger values.

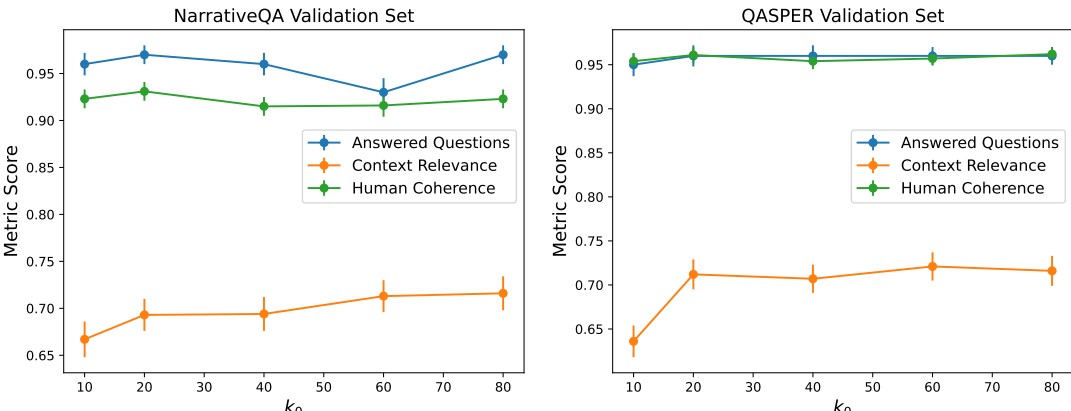

Figure 12: Comparison of different values of $k_0$ for the postQFRAP algorithm on NarrativeQA and QASPER validation sets.

## F    QUERY EXPANSION

A possible extension to the postQFRAP algorithm is to first expand the query $q$ before retrieving $k_0$ initial documents. This would lead to broader documents being retrieved before the clustering-summarization process, potentially leading to an improved context.

To test this approach, we use the query expansion algorithm from (Jagerman et al., 2023) with the Q2E/PRF prompt. It consists of first retrieving the top-3 documents using naïve RAG. Then, we ask an LLM to extract key words from those documents relevant to the question. We append those to the query that is duplicated 5 times to get the augmented query $q'$:

$$q' = \text{Concat}(q, q, q, q, q, \text{LLM}(\text{prompt}_q))$$

where $\text{LLM}(\text{prompt}_q)$ is the output of the Q2E/PRF prompt. The latter is found in Table 9. Then we use postQFRAP as before, by replacing $q$ with the augmented query $q'$.

We use this particular query expansion algorithm instead of the simpler Q2D/ZS algorithm that just asks an LLM to answer the query and use that answer as a new query because we do not want to exploit the LLM's parametric knowledge. Instead, we ground the extension on the retrieved documents.

We present the results in Figures 13 and 14. We observe that adding Query Expansion to postQFRAP does not improve performance and, in fact, reduces the comprehensiveness and coherence of the generated answers.

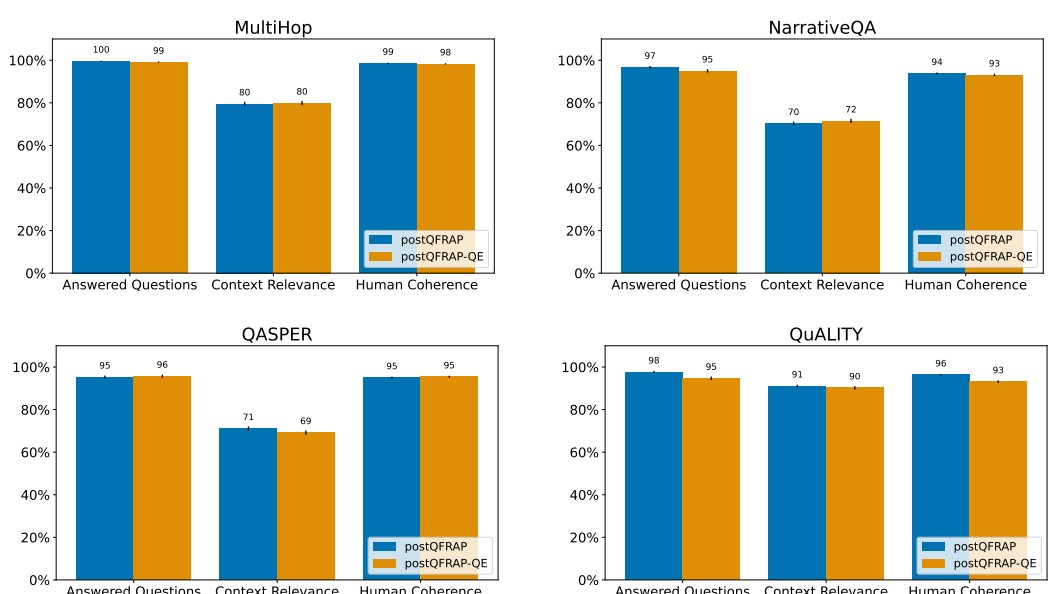

Figure 13: Comparing postQFRAP vs postQFRAP with Query Expansion on 4 datasets.

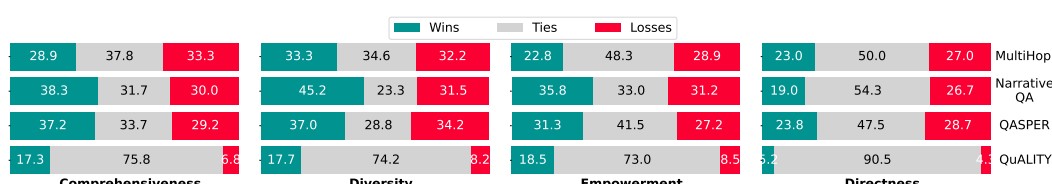

Figure 14: Percentage of Wins, Ties and Losses for postQFRAP (green) vs postQFRAP with Query-Expansion (red) on 4 datasets.

## G  PROMPTS

Table 3: Prompt for Question Generation

| Role | Content |
|---|---|
| system | Given some text, you generate questions that can be answered by that text. |
| user | You are given a context and a few example questions. Your task is to thoroughly contemplate these excerpts and conceive one question or query that a human with interest in the subject matter might pose, ensuring that the answer to that question can likely be found within the provided text segment. |
| | You should not use general references like "What documents do I need to submit to demonstrate compliance with this directive?" or "What are the requirements for compliance with this directive?". Instead, you should use specific references like "What documents do I need to submit to demonstrate compliance with the deposit guarantee schemes directive?". Do not mention the context in the question. If you are unable to generate a high-quality question, return IMPOSSIBLE instead. |
| | Example of questions: {questions} |
| | Context: {context}: |

Table 4: Prompt for Text Summarization

| Role | Content |
|---|---|
| system | You are a helpful assistant. |
| user | Write a summary of the following, including as many key details as possible using at most {max_tokens} tokens: |
| | {context} |

Table 5: Prompt for Query-Focused Text Summarization

| Role | Content |
|---|---|
| system | You are a helpful assistant. |
| user | Summarize the information in the retrieved documents using at most {max_tokens} tokens. Make sure to include in your summary all the details that can be used to answer the question and omit any details that are entirely irrelevant to the question. |
| | Retrieved documents: {context} |
| | Question: {question} |
| | Summary: |

Table 6: Prompt for Computing Human Coherence Rating

| Role | Content |
|---|---|
| system | You are a helpful assistant. |
| user | You are given a question and an answer. Your task is to evaluate whether the provided answer could have been generated by a human expert, focusing on the coherence of the response. Assess how logically and smoothly the ideas are connected, how well the answer flows, and whether it maintains a clear and consistent structure. Provide a brief explanation of your reasoning, and then rate the likelihood on a scale of 1 to 5, where: |
| | 1: Very unlikely to have been generated by a human expert (e.g., disjointed or lacking logical flow) |
| | 2: Unlikely (e.g., partially coherent but ideas do not flow well or seem disconnected) |
| | 3: Possibly (e.g., somewhat coherent but with noticeable breaks in flow or structure) |
| | 4: Likely (e.g., mostly coherent with minor disruptions in flow or structure) |
| | 5: Very likely to have been generated by a human expert (e.g., highly coherent, logically structured, and well-organized). |
| | The final line of your output must be an integer between 1 and 5. |
| | Question: {question} |
| | Answer: {answer} |

Table 7: Prompt for Question Answering

| Role | Content |
|------|---------|
| system | You are a Question Answering Portal. Given a question with relevant information sources, your task is to respond to the question using ONLY information from the provided sources. Ensure that the facts included are directly related to answering the question. If the sources do not provide an answer, reply with "No information is provided in the sources." |
| user | Sources: {context}
Question: {question}
Generate an answer with at most {max_tokens} tokens.
Answer: |

Table 8: Prompt for One-Shot Context Summarization (Zhang et al., 2024)

| Role | Content |
|------|---------|
| system | You are a helpful assistant. |
| user | Instruction: You will be given a query and a set of documents. Your task is to generate an informative, fluent, and accurate query-focused summary. To do so, you should obtain a query-focused summary step by step.
Step 1: Query-Relevant Information Identification
In this step, you will be given a query and a set of documents. Your task is to find and identify query-relevant information from each document. This relevant information can be at any level, such as phrases, sentences, or paragraphs.
Step 2: Controllable Summarization
In this step, you should take the query and query-relevant information obtained from Step 1 as inputs. Your task is to summarize this information. The summary should be concise, include only non-redundant, query-relevant evidence. The output summary must consist of at most {max_tokens} tokens.
Query: {question}
Documents: {context} |

Table 9: Prompt for Q2D/PRF Query Expansion

| Role | Content |
|------|---------|
| system | You are a helpful assistant. |
| user | Write a list of keywords for the given question based on the following context. Use at most {max_tokens} tokens:
Sources: {context}
Question: {question}
Keywords: |

# H   DETAILED EXPERIMENTS

## H.1   COMPUTING RESOURCES

We implement our algorithms in Python 3.10 and run our experiments on a standard laptop with 16GB of RAM and 12th Gen Intel(R) Core(TM) i7-1250U CPU.

## H.2   adRAP RUNTIME

We present in Table 10 the time taken and the number of summary calls made on each of the QASPER and QuALITY dataset for two different algorithms:

- Build the full tree on the first 70% of the dataset, then use the adRAP algorithm to add the remaining 30%.
- Build the full tree on the first 70% of the dataset, then re-compute the full tree from scratch on the full dataset.

Table 10: Comparison of time taken and the number of summary calls between building a tree on 70% of the dataset followed by adRAP, and computing the full tree twice: once on 70% of the dataset and again on the entire dataset.

| Dataset | Time Taken (adRAP Algorithm) | Time Taken (Full Tree Computed Twice) | Summary Calls (adRAP Algorithm) | Summary Calls (Full Tree Computed Twice) |
|---------|---------|---------|---------|---------|
| QASPER | 638 s | 1,093 s | 530 | 761 |
| QuALITY | 342 s | 524 s | 372 | 451 |

It is clear that using adRAP requires significantly less time and fewer summary calls compared to re-computing the full tree, even when the latter is done only once. If the full tree were to be re-computed each time a new document is added, the difference would become substantially larger.

## H.3   GENERATING QUESTIONS

To create more challenging questions that require a broader understanding of the dataset, we take the following approach. First, we construct a RAPTOR tree on top of the dataset. Then, to generate a new question, we sample a node from the tree and prompt a LLM to create a question based on the text from that node. We provide the LLM a few high quality questions to improve its output. The key idea is that some RAPTOR nodes contain summaries of various chunks, meaning the generated question requires synthesizing and summarizing information from different documents to be answered. The prompt used for generating these questions can be found in Table 3.

## H.4   ADDITIONAL DATASET: *NarrativeQA*

*NarrativeQA* consists of complete stories and questions designed to assess a deep, comprehensive understanding of the narratives (Kočiský et al., 2017). From this dataset, we select the first 300 questions along with their corresponding documents.

Figures 15 and 16 show that, on the NarrativeQA dataset, adRAP performs comparably to RAPTOR and Greedy adRAP, while consistently outperforming Naïve RAG.

Figure 17 demonstrates that query-focused algorithms clearly outperform the baselines on the NarrativeQA dataset. Notably, postQFRAP and one-shot summarization achieve comparable results. However, as shown in Figure 18, postQFRAP continues to significantly outperform all other algorithms in terms of comprehensiveness, diversity, and empowerment of the generated answers.

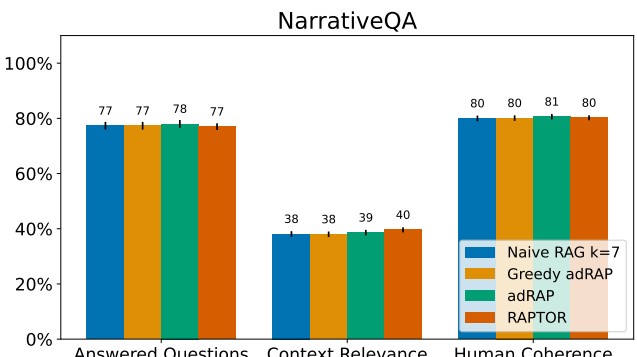

Figure 15: Evaluation of adRAP (Section 4.4) vs other algorithms on NarrativeQA.

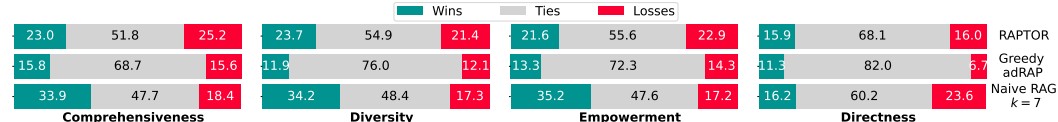

Figure 16: Percentage of Wins, Ties and Losses for adRAP vs other algorithms on NarrativeQA.

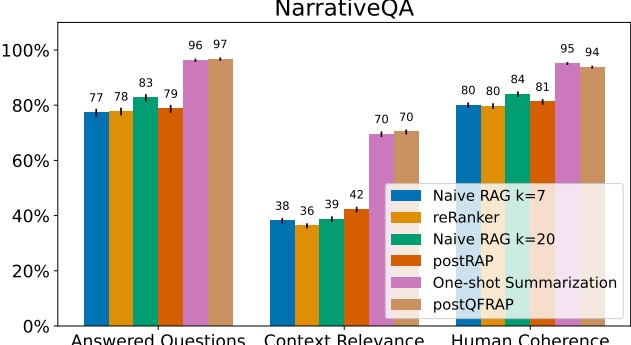

Figure 17: Evaluation of postQFRAP (Algorithm 3) vs other algorithms on NarrativeQA.

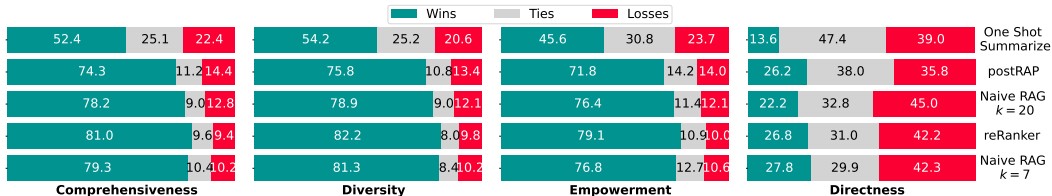

Figure 18: Percentage of Wins, Ties and Losses for postQFRAP vs other algorithms on NarrativeQA.

## H.5 DATASET PREPROCESSING

For datasets with HTML symbols (NarrativeQA and QuALITY), we use the BeautifulSoup [9] library to convert them to text. For all datasets, we standardize the formatting by cleaning new lines, ensuring that only two new lines separate different paragraphs.

## H.6 DATASETS STATISTICS

In Table 11, we present the sizes of the datasets used in our experiments. In Table 12, we present various statistics regarding the recursive-abstractive trees constructed from our datasets. The number of internal nodes is approximately $n/6$, where $n$ represents the total number of chunks in the dataset. Additionally, most cluster sizes range between 4 and 15, with nodes rarely belonging to more than one cluster.

| Dataset | Number of Tokens | Number of Questions |
|---|---|---|
| MultiHop | 1,394,859 | 230 |
| NarrativeQA | 939,474 | 300 |
| QASPER | 364,610 | 300 |
| QuALITY | 254,297 | 300 |

Table 11: Sizes of the used datasets

| Dataset | Number of Leaves | Number of Internal Nodes | Cluster Size | Number of parents per leaf |
|---|---|---|---|---|
| MultiHop | 6,489 | 935 | $7.96 \pm 4.85$ | $1.004 \pm 0.063$ |
| NarrativeQA | 4,083 | 499 | $9.45 \pm 5.81$ | $1.034 \pm 0.18$ |
| QASPER | 2,072 | 462 | $5.487 \pm 1.96$ | $1.002 \pm 0.044$ |
| QuALITY | 1,064 | 250 | $5.24 \pm 2.11$ | $1.0 \pm 0.0$ |

Table 12: Statistics of the different recursive-abstractive trees we construct on the datasets

---

[9] https://www.crummy.com/software/BeautifulSoup/bs4/doc/

