# OpenReview forum: "Recursive Abstractive Processing for Retrieval in Dynamic Datasets"
_ICLR.cc/2025/Conference — Submitted to ICLR 2025_

### Official Review · Reviewer_PdoE · 2024-11-03

**Soundness:** 2
**Presentation:** 2
**Contribution:** 1
**Rating:** 3
**Confidence:** 3

**Summary:**

RAPTOR is inefficient in dynamic setting where index needs to be frequently updated as it requires recomputing clusters and summaries in the hierarchical tree. To address this limitation, this paper proposes ADRAP and POSTQFRAP algorithms. ADRAP is used to update the tree without full recomputation by recalculating clusters and regenerating summaries only for nodes directly affected by the addition or removal of documents. POSTQFRAP retrieves a larger pool of documents and builds recursive-abstractive tree. It uses query-focussed summarization when building summary nodes.

Authors evaluate both methods on MultiHop, QASPER and QuALITY datasets. Authors used following metrics for evaluation: 1. answered question, 2. context relevance, 3. Human coherence metrics. In addition, authors used win-rate on comprehensiveness, diversity, empowerment, and directness.

**Strengths:**

- ADRAP addresses the most important limitation of the RAPTOR method. It is computationally efficient. As shown by Table 10, it's faster and requires fewer summarization calls than recomputing entire tree with RAPTOR.

- Both models have performance close to RAPTOR.

**Weaknesses:**

- Evaluation Metrics:

Authors have used indirect metrics for evaluations. LLM Judge is not the most reliable evaluation protocol if it is possible to perform direct comparisons. You should consider standard metrics like accuracy (QuALITY)/ F1 (QASPER) for better understanding.

- Baselines:

ADRAP- It would be useful to include two baselines 1. where we do not update the RAPTOR tree and just retrieve from old RAPTOR tree and new documents, 2. construct a new raptor tree only on new documents and retrieve from new + old raptor trees.

postQFRAP- a version of the model without explicit clustering (e.g., Tree Summarize, Create and Refine, Accumulateresponse synthesizers in LlamaIndex)

- Datasets:

Both QuALITY and QASPER datasets include question that can be answered by a single document. Do you think they make a suitable dataset for your evaluation? Original RAPTOR work built one tree per document and answer questions based on that. However the benefit of your approach only comes when we know that building one tree for all documents is required. Otherwise, we can just build one tree per document and retrieve from that.
In my opinion, you should consider evaluating your approach on complex questions that require multi-document evidence.

**Questions:**

- Why didn't you use standard metrics for evaluation? MultiHOP can be objectively answered, and there are established metrics/ evaluation protocol for other two datasets as well.

- Refer to question about datasets in Weakness.

---

> ### Author Response · Authors · 2024-11-20
>
> Dear Reviewer PdoE, we thank you for your time in evaluating our submission and we are grateful for your comments. Please find below responses to the questions raised in your reviews.
>
> *Evaluation metrics*: For MultiHop, we generated our own questions from the RAPTOR tree to simulate more challenging questions. This means there is no ground-truth, and thus we used comparison-based metrics. We do acknowledge that presenting accuracy/F1 for the other datasets would strengthen our experiments.
>
> *Baselines*: We thank you for the suggestions, and plan on incorporating them in a future revision.
>
> *Datasets*: We generated questions for MultiHop to get more interesting examples as you mentioned. The QuALITY and QASPER datasets were left unchanged since they gave us results that clearly differentiate adRAP/RAPTOR vs greedy/RAG as well as postQFRAP vs non-QFS approaches.

---

> > ### Comment · Reviewer_PdoE · 2024-11-28
> >
> > Thank you for the rebuttal, I will keep my score unchanged, please incorporate any reviewer suggestions that you believe would strengthen the paper.

---

### Official Review · Reviewer_aHiU · 2024-11-04

**Soundness:** 2
**Presentation:** 1
**Contribution:** 1
**Rating:** 3
**Confidence:** 3

**Summary:**

This paper aims at handling dynamic data and improving retrieval performance for RAG applications. It proposes adRAP, an extension of an existing work (RAPTOR), to efficiently approximate clustering when documents are added or removed. It also proposes postQFRAP, a post-retrieval algorithm that applies query-focused and recursive-abstractive processing to refine large contexts.

**Strengths:**

1. The method presented by the authors is mathematically interesting and seems sound (equation 3).

2. I like the pseudo-code written by the authors, which improves clarity of the paper.

**Weaknesses:**

1. I am not sure whether dynamic data for RAG is a high-impact research problem. First of all, building a tree for the entire dataset is a one-shot process, so I think the efficiency bottleneck is the inference speed. Secondly, if we just add a small number of documents, we can simply assign those documents to existing clusters (your way of adapting RAPTOR) or create a new cluster to hold these newly added documents. I guess the performance will not be affected significantly due to the small number of documents added (your experiment demonstrates this point). If we expect that most of the queries are related to the small number of documents we add, I think we can just restart RAPTOR and build a tree dedicated for our updated dataset. If we want to add a lot of documents, you mention that a full tree recomputation is still needed. So I am not sure the contribution of this paper.

2. For postQFRAP, you claim to build a tree for all retrieved chunks, which sounds quite inefficient. As multiple summarization calls are required to build a tree for each query, this design annihilates your original purpose of being efficient for a small number of dynamic data.

3. Performance is weak. As discussed in the previous two points, it seems that "adRAP + postQFRAP" has similar performance as RAPTOR. Efficiency-wise, I am not sure whether it would be more efficient to use "adRAP + postQFRAP", as I mention above that you are using a more inefficient retrieval method than methods mentioned in RAPTOR (e.g., SBERT and DPR). The most salient improvement I see is "adRAP exceeds RAPTOR in metrics such as comprehensiveness, diversity, and empowerment" on QuALITY as described in line 443, but I am not sure whether the improvement comes from adRAP or postQFRAP.

4. It is questionable whether the proposed method can be applied on other state-of-the-art baselines such as GraphRAG ("From Local to Global: A Graph RAG Approach to Query-Focused Summarization"), so the impact of this paper might be very limited.

5. Experiment is limited. There is no ablation study about the effectiveness and efficiency of either adRAP or postQFRAP.

6. Writing is verbose in some parts of the paper. For example, section 3 is too long with many unnecessary details, since you are essentially providing the background of RAG and existing work in around 2 pages. There are many basic things out there that can be omitted (e.g., the EM algorithm). I suggest to simplify this section, and you just need to cite the related work with short explanation. Another option is to put those details into appendix. On the other hand, I suggest to put more words for the last paragraph of section 1 to improve clarity (otherwise, reviewers need to read the whole paper to be able to tell). For example, how much does "postQFRAP effectively enhances retrieval quality"? Are you using state-of-the-art baselines? What are those "real-world datasets"?

**Questions:**

1. Have you done any detailed analysis about the overall efficiency between "adRAP + postQFRAP" and RAPTOR when you only aim to add a small number of documents?

---

> ### Author Response · Authors · 2024-11-20
>
> Dear Reviewer aHiU, we thank you for your time in evaluating our submission and we are grateful for your comments. Please find below responses to the questions raised in your reviews.
>
> 1 - When we expect many similar documents to be added, we could indeed build another RAPTOR tree. However, we would lose the potential connections between these documents and other slightly less similar documents that would remain in the initial RAPTOR tree. Instead, with adRAP, we update the RAPTOR tree so that the summaries are maintained at multiple levels.
>
> 2 - We would like to clarify that postQFRAP was not designed as an addition to adRAP, rather as an alternative to avoid the development burden of maintaining the processed dataset. This indeed shifts the computational burden to retrieval time.
>
> 3 - Kindly note that our experiments are for adRAP only, then postQFRAP only, we did not try “adRAP + postQFRAP”. Our goal is to show that we get similar performance to RAPTOR, the target model we are approximating with adRAP.
>
> 4 - adRAP is indeed focused on RAPTOR, although we believe a very similar approach can be used with clustering-based GraphRAG. postQFRAP on the other hand is flexible and can be directly used with any retrieval algorithm.
>
> 5 - Our experiments compared adRAP with a greedy variant that did not split clusters. For postQFRAP, we compare it against a non-query-focused variant (postRAP). We agree that this can be made more explicit, and plan on adding further ablation studies.
>
> 6 - We thank you for this suggestion, and plan to improve this section in later revisions.
>
>
> *“Have you done any detailed analysis about the overall efficiency between "adRAP + postQFRAP" and RAPTOR when you only aim to add a small number of documents?”*
>
> We did not compare adRAP with RAPTOR for a small number of documents, as we believe the results will be very similar for both approaches. We thus focused on a more challenging setting where more documents are added.

---

> > ### Comment · Reviewer_aHiU · 2024-11-27
> >
> > Thanks for your rebuttal! However, my concerns are not addressed and I cannot increase my scores. For example, it seems that your adRAP and postQFRAP are designed to work separately with different goals. Then it sounds like two different papers, because I need more details to understand your motivation, method, etc. Moreover, check out the weakness 1 I wrote. I suggest to respond to each of my point in a clearer way.

---

### Official Review · Reviewer_6eaf · 2024-11-05

**Soundness:** 2
**Presentation:** 2
**Contribution:** 2
**Rating:** 3
**Confidence:** 4

**Summary:**

This paper addresses the dynamic document challenge problem in RAG for recursive abstractive indexing of a corpus. Specifically, to efficiently maintain the recursive abstractive indexing tree, the authors propose adRAP,  an algorithm which only partially update indexing trees when new documents are added. Furthermore, a post-retrieval summarization algorithm is also proposed -- postQFRAP, which builds an online query-focused recursive-abstractive tree for more relevant context selection. Experimental results show that the proposed method can achieve competitive performance with fully offline tree indexing algorithms such as RAPTOR baseline.

**Strengths:**

In general, document updating is a very common problem in document indexing, therefore it is useful to develop online indexing algorithms which can effectively address the dynamic dataset problem. And online clustering algorithms are reasonable solutions for this problem.

**Weaknesses:**

My main concerns about this paper are as follows:
1. The contribution is incremental, the proposed method is an extension of previous recursive abstractive indexing algorithms;
2. The two contributions, adRAP and postQFRAP, are separated. adRAP addresses the dynamic dataset problem and postQFRAP addresses the online organization problem of retrieved documents. The paper should focus on one main contribution;
3. The proposed adRAP algorithm is just an online clustering algorithm with tree structure updating strategy. Therefore I think the authors should explain and compare different online clustering algorithms. In my opinion, the proposed  problem can be better addressed using the theory-guided Chinese restaurant process, rather than the current ad hoc clustering algorithm.
4. The baselines are too weak. I think the authors should also compare different online clustering algorithms, and different online search result organization algorithms in current RAG studies.
5. The current presentation of experimental results is hard to follow and analyze.

**Questions:**

See Weaknesses in above.

---

> ### Author Response · Authors · 2024-11-20
>
> Dear Reviewer 6eaf, we thank you for your time in evaluating our submission and we are grateful for your comments. Please find below responses to the questions raised in your reviews.
>
> 1 - adRAP is indeed an extension of RAPTOR. postQFRAP on the other hand is a novel post-retrieval algorithm.
>
> 2 - We acknowledge this and plan to focus on postQFRAP in a future revision, as it is a novel post-retrieval algorithm that achieves state-of-the-art results.
>
> 3 - We did not include an extensive comparison with other online clustering algorithms, as our setting differs in that we initially have a GMM fit with access to all of the points and we want to minimize the number of updated clusters. Previous work either assumes that previous points are not accessible, or does not try to optimize on the number of updated clusters.
>
> 4 - We acknowledge that including other online algorithms would be helpful. As for post-retrieval methods, we compare our algorithm with state-of-the-art approaches (re-ranking and query-focused-summarization).
>
> 5 - We plan on improving the presentation in a future revision of the paper.

---

> > ### Comment · Reviewer_6eaf · 2024-11-28
> >
> > Thanks for your rebuttal!  Although I I think my concerns are still here, but I think the authors could improve their paper based on the comments of all reviewers.

---

### Meta-Review · Area_Chair_PpPe · 2024-12-10

**Metareview:**

This paper addresses the challenge of updating retrieval corpora for  RAG systems when new documents are available . The authors  propose adRAP for efficiently updating the recursive abstractive indexing tree, and postQFRAP, a post-retrieval algorithm that builds a query-focused tree for improved context selection.

While reviewers acknowledge the importance of the problem, they highlight several concerns regarding the method and evaluation:

1.	Limited Novelty: adRAP can be viewed as a standard online clustering algorithm, thus the novelty is rather questionable.
2.	Evaluation Metrics: Reliance on LLM Judge is questioned, with suggestions to include standard metrics like accuracy and F1.
3.	Weak Baselines: The comparison includes only relatively weak baselines, limiting the strength of the evaluation.

Overall, while the problem is important, the proposed method and evaluation are not strong enough for an acceptance in ICLR. The recommendation is a rejection.

**Additional Comments On Reviewer Discussion:**

The reviewers highlighted concerns regarding limited novelty, the need for additional evaluation metrics, and stronger baselines, as noted in the previous section. Following the rebuttal, they suggested that further experiments and clearer explanations were required, but these points were not fully addressed in the response.

---

### Decision · Program_Chairs · 2025-01-22

Reject